# Associations between Parent–Child Communication on Sexual Health and Drug Use and Use of Drugs during Sex among Urban Black Youth

**DOI:** 10.3390/ijerph18105170

**Published:** 2021-05-13

**Authors:** Donte T. Boyd, Ijeoma Opara, Camille R. Quinn, Bernadine Waller, S. Raquel Ramos, Dustin T. Duncan

**Affiliations:** 1College of Social Work, The Ohio State University, Columbus, OH 43210, USA; Quinn.395@osu.edu; 2Center for Interdisciplinary Research on AIDS (CIRA) at Yale University, New Haven, CT 06510, USA; Ijeoma.opara@yale.edu; 3Department of Social and Behavioral Sciences, School of Public Health, Yale University, New Haven, CT 06510, USA; 4School of Social Work, Adelphi University, Garden City, NY 11530, USA; bwaller@adelphi.edu; 5Rory Meyers College of Nursing, New York University, New York, NY 10010, USA; Raquel.ramos@nyu.edu; 6Department of Epidemiology, Mailman School of Public Health, Columbia University, New York, NY 10032, USA; dd3018@cumc.columbia.edu

**Keywords:** drug use, protective mechanisms, Black youth, parenting, sex

## Abstract

Black youth and their families living in urban settings may experience unique stressors that contribute to underlying issues due to the environmental context. Such factors may exacerbate and promote drug use and engagement in risky sexual behaviors, unknowingly. Little is known about how family factors, peer pressure, condom use, and other related factors are associated with substance use and engaging in sexual behaviors while on drugs among urban African American youth aged 12–22 (N = 638). We used regression models to examine associations between parental bonding, parent–adolescent sexual health communication, condom use, peer pressure on substance use, and having sex while on drugs. Multivariate results indicated that parental bonding was statistically significant and associated with drug use (OR: 1.36, 95%CI: 1.36). Our study highlights that parental bonding plays a critical role in youth using drugs while living in urban environments.

## 1. Introduction

There is a dearth of literature that has indicated that substance and alcohol abuse is a public health concern among the youth in the United States [1,2]. According to the National Survey on Drug Use, approximately 22.5 million Americans aged 12 years and older were illicit drug users (i.e., they used drugs in the previous 30 days from when the survey was conducted), one-quarter had reported recent binge alcohol use, and approximately 6% had reported frequent binge drinking [3,4,5]. Prior research has reported that approximately 20% of the youth had tried illicit drugs by the time they were in eighth grade, and the percentage steadily increased to 50% by the time they reached 12th grade [6]. Alcohol use has been reported to be widespread during this stage of youth, with approximately 70% reporting that they tried alcohol during this stage, 25% reporting being drunk, and 32.8% reporting they had at least one drink of alcohol within the past 30 days [4,5,7,8]. It is critical to understand alcohol and substance use, as it has contributed to poor sexual health outcomes among urban youth [8].

For Black youth, alcohol use is far more complex than it is for their White peers and other racial and ethnic groups [2,8]. Black youth are more likely to start drinking at a later age than their peers and consume less alcohol [5,9]. However, Black youth experience more negative social consequences from drinking, report more alcohol-related illnesses and injuries, and are less likely to recover from alcohol dependence when compared to White youth [2,10,11]. Although the rate of alcohol use is significantly lower among Black youth compared to other racial and ethnic groups, the rate of illicit drug use among them is higher [5]. A plethora of studies have focused on the impact of substance use on sexual health in urban communities [12,13]. However, very few studies have focused on the role of the Black family and their impact on the sexual risk behaviors and alcohol and substance use among Black youth.

Those in urban centers are especially at risk of negative sexual outcomes for multiple reasons, including greater exposure to sexually transmitted infections (STIs) [14,15]. Black youth residing in urban areas typically reflect lower rates of substance use and co-occurring sexual activity compared to their White peers; however, they experience poorer health outcomes, including elevated rates of contracting HIV and STIs [16,17,18,19,20,21]. While drug use rates among Black youth are lower than those among other racial groups, Black youth who use drugs tend to have worse health outcomes and negative consequences associated with drug use, such as suspension from school, involvement in the (juvenile) criminal justice system, and poor sexual health outcomes [22,23]. While prior research has primarily focused on risk factors for Black youth, there is emerging evidence regarding protective factors that serve as a buffer against substance and drug use for Black youth living in the urban environment.

Parent–child communication has been shown to be a significant protective factor for youth [21,24], and evidence has indicated that parent–child communication is the preferred source of quality sexual health education as well as an effective means of maintaining sexual well-being among urban Black youth [25,26]. Black male youth residing in the inner city whose mothers are supportive and openly communicate with them about their sexual health are less likely to take sexual risks that lead to unprotected sex, unwanted pregnancies, and HIV and other STIs [27,28,29]. Mothers are typically the lead communicators in Black families and generally assume the role of the sexual health educator within the home [29]. Sexual health conversations with parents are also positively correlated with urban Black males’ improved decision making regarding sexual health [29].

Some promising, albeit limited evidence has revealed the effects of Black fathers taking the lead in facilitating sexual health communication with their children. Black youth in urban settings whose fathers freely converse with them about the responsibility of sexual intercourse generally abstain, delay initial sexual experiences, and employ protective measures when they do decide to become sexually active [30,31]. Harris et al. [28] utilized a sample of 100 father–son dyads in an urban setting and found that emotional connectedness and open communication between fathers and sons resulted in youth taking fewer sexual risks along with an increased likelihood that they would practice safe sex.

### 1.1. Lack of Research on Sexual Health and Drug Use among Black Youth (Including Research on Parent–Child Communication)

Ecological and cultural factors influence Black youth’s engagement in sexual risk behaviors and drug use [32]. At the ecological level, family and other social and contextual factors (i.e., gender, social class, etc.) impact sexual risk behaviors and drug use among this population. For instance, family substance use history and attitudes, family conflict, lack of family support, and high parent–child conflict are factors that contribute to substance use and the youth’s engagement in sexual risk behaviors [19,33,34,35]. Other literature has demonstrated that youth are more likely to use condoms, have reduced numbers of sexual partners, get tested for HIV, and use drugs when they have positive relationships with their parents and communicate with their parents about sex [19,21,24]. In addition, prior research has demonstrated that drug use often co-occurs with sexual intercourse that is influenced by these ecological factors. However, there is a gap in the literature on how Black parents influence their teenage children through communication on sex and bonding and how these factors influence whether their child will have sex under the influence of drugs or alcohol [36].

### 1.2. Contextual Risk Factors among Urban Black Youth

Urban Black youth who live in under-resourced neighborhoods often experience poverty and are exposed to high crime rates and disorganized communities. This exposure may indirectly make them predisposed to a sense of hopelessness about their future [19]. Youth research has found that hopelessness is associated with having sex at an early age, multiple sexual partners, and condomless sex [37,38]. Context is especially important because adolescence is a period of time when youth are mostly influenced by peers in their immediate surroundings where social norms are developed, maintained, communicated, and replicated [38]. Consequently, it is essential to understand Black families’ parenting practices and peers influence for a more in-depth understanding of its effects on the youth’s likelihood of engaging in risk-taking behaviors.

### 1.3. Ecodevelopment Theory

The theory of ecodevelopment provides a comprehensive understanding of how Black urban youth’s socio-contextual position and the interplay of these factors with their development elucidate the risk and protective factors that operate during adolescence [39]. Based on Bronfenbrenner’s construction of social ecology, the ecodevelopment theory postulates that youth development is dependent on influences from four interrelated systems: the microsystem (i.e., parental conversations on sexual health and drug use), the mesosystem (i.e., how peers are monitored by their parents), the exosystem (i.e., parental support systems), and the macrosystem (i.e., culture and cultural shifts) in which they are situated [40,41]. Additionally, the ecodevelopment theory posits that youth’s interactions with and the ways in which they are influenced by these external systems influence their perspectives and behaviors and shape the corresponding risk and/or protective factors as well as their potential relationships. This theory reflects many of the ways in which urban Black youths’ socio-cultural context may influence their behaviors, including risky behaviors. Further, it is important to center adolescent health outcomes as a result of social positions across these systems based on parent–child communication as a family mechanism [42].

### 1.4. Current Study

While the misuse of alcohol and drugs is a growing problem in the United States, it has become a serious problem for youth and young adults in urban areas. Although substance misuse can occur at any age, the adolescent and young adult years are particularly critical at-risk periods. Prior research has indicated that youth who have used substances are more likely to engage in sexual risk behaviors, such as becoming sexually active, having multiple sex partners, not using a condom, and undergoing pregnancy before the age of 15. To understand Black youth’s experience with substance, use and sex under the influence of drugs or alcohol, we examined the role of parent–youth bonding and sexual health communication and investigated whether contextual factors contribute to or prevent substance use and sex while on drugs among this population in an urban context. This study drew on the ecodevelopment theory, which was a fitting theoretical lens that was employed to interrogate the effect of parent–child communication on sexual health as a protective factor against urban Black youth’s likelihood of engaging in illicit drug use and co-occurring sexual risk behaviors.

## 2. Methods

### 2.1. Sampling

Data were collected from the Resilience Project of 2014, a study that examined protective and risk factors related to the sexual risk behaviors of urban Black youth living in large Midwestern cities [8,9]. Participants were recruited from high schools, youth programs and church groups, and public spaces where the youth congregate in the community (e.g., movie theaters and parks). The majority of the participants lived in predominately Black and low-income urban neighborhoods, and the average household income was between USD 24,049 and USD 35,946, which was below the city average of approximately USD 43,000. Based on the 753 participants who were initially invited to enroll in the study, the response rate for this study was 87%. Youth were eligible for the study if they identified as Black/African American and were between the ages of 12 and 22.

### 2.2. Procedures

For participants under the age of 18, parental consent and youth assent were obtained for all study participants, and a meticulous letter describing the study was provided to parents. To collect data from schools, youth programs, and church groups, researchers received permission to recruit participants from principals and group leaders. Flyers describing the study were posted at each of these locations, and trained research assistants introduced the study to potential participants in these settings. All research assistants completed human subjects training, which included collecting informed consent and protecting the rights and confidentiality of each participant. Only youth who returned signed consent forms participated in the study. Youth who were recruited in public venues were only allowed to participate in the study if their parent was present and provided consent.

Participants (N = 638) who were recruited from community programs, churches, and schools received the questionnaire in these respective locations. For those who were recruited in public venues, the questionnaires were given in quiet spaces, such as a library, at or near those venues, which were chosen by the participant. Approximately 45 min were required to complete the questionnaire, and the youth were compensated with a USD 10 gift card. This study was approved by the Institutional Review Board.

### 2.3. Measures

The two outcome variables of this study were drug use and sex under the influence of drugs or alcohol. Drug use was assessed by the use of three or more recreational drugs: “Have you taken a substance such as cigarettes, ecstasy, codeine, alcohol, and marijuana in the last 30 days?” Sex while on drugs or alcohol was assessed using the following question: “The last time you had sexual intercourse, did you have any alcoholic drinks and/or take any drugs before having sexual intercourse?” These dichotomous (i.e., yes and no) variables were derived from previous studies [8,9]. Associations between these outcomes and contextual and sociocultural variables were examined.

Several contextual variables were collected and included as covariates. Participants were asked to indicate their age, gender, sexual orientation, and whether they were receiving government assistance [8,9]. Participants were also asked the following questions: “In the past 12 months, have you had sex with someone in exchange for something other than drugs?” and “In the past 12 months, have you had sex with someone in exchange for drugs?” Both were dichotomous variables with yes or no responses. Condom use was assessed using the following question: “The last time you had sexual intercourse, did you use a condom?”, and this was based on a dichotomous (i.e., yes and no) response.

Several measures were used to assess the cultural variables.

### 2.4. Parent–Adolescent Sexual Health Communication

This was measured using a 12-item scale, and the respondents were asked about how much information their parents had shared with them. Participants responded to items such as “human sexuality (with human sexuality defined as what sex is, how to have sex, and why people have sex)”, “menstruation”, and “HIV/AIDs”. The response categories ranged from 1 = none to 5 = extensive, and higher scores indicated a higher degree of communication between the youth and their parent regarding sexual health. The Cronbach’s alpha for this scale was 0.92 (M = 22.82, SD = 12.65, range = 0–48).

### 2.5. Parental Bonding

This four-item scale (ranging from 1 = not at all to 5 = very much) asked the respondents the following questions: (1) “How close do you feel to your father?”; (2) “How close do you feel to your mother?”; (3) “How much do you think your father cares about you?”; (4) “How much do you think your mother cares about you?” These items were reverse-scored as necessary such that a higher score indicated more of the attribute named in the label. The Cronbach’s alpha for this scale was 0.76 (M = 3.85, SD = 1.064, range = 0–5).

### 2.6. Peer Pressure

This was measured using a 12-item scale, and the respondents were asked about their 10 closest friends, i.e., persons they trusted and depended on and who influenced their ideas and behaviors. Participants responded to items such as “How many of your 10 closest friends drink alcohol?” and “How many of your 10 closest friends have smoked marijuana?” The response categories ranged from 1 = none to 5 = most, and higher scores indicated an increase in the number of peers influencing their ideas and behavior. The Cronbach’s alpha for this scale was 0.90 (M = 1.15, SD = 0.86, Range = 0–4).

### 2.7. Statistical Analysis

All analyses were performed using STATA 16. All analyses were conducted on observations that included non-missing data for both outcome variables—Substance use and sex while on drugs. Statistical tests of associations were conducted between the independent variables (peer pressure, condom use, parental bonding, and parent–adolescent sexual communication) and the covariates (age, gender, sexual orientation, government assistance, having sex in exchange for drugs, and having sex in exchange for something other than drugs) the two outcome variables, drug use and sex while on drugs. We conducted univariate analyses to describe the overall sample. Additionally, we conducted Chi-square tests test for binary and categorical measures between the study outcomes (i.e., drug use and sex while on drugs or alcohol use) and the contextual variables. Next, prevalence ratios (PR) were generated using a Poisson regression with robust errors due to high prevalence of the outcome, drug use [43,44,45]. The model was used to estimate the adjusted prevalence ratios, controlling for potential confounders selected a priori based on the prior literature. Lastly, a multivariate analysis was conducted on the outcome sex under the influence of drugs or alcohol, and their adjusted odds ratios, their 95% confidence intervals, and respective *p*-values were calculated. Survey data were analyzed based on listwise deletion, and for the survey scales, a mean score of the scale items was generated for participants with non-missing data.

## 3. Results

### 3.1. Sample Characteristics

The analytic sample for this study was composed of 636 participants. Table 1 displays the descriptive statistics of the overall sample relative to all of the major variables of the study. Among the participants, 45% were male and 54% were female, and the mean age was 16 years (SD = 1.41). Slightly over three-fourths (76.5%) of the overall sample qualified for free or reduced school lunch. Approximately 60% of the sample stated that they had made their sexual debut.

Of the 636 Black youths, 60% (372) reported the use of marijuana in the past three months. Approximately 49.5% (315) of youths reported using alcohol in the past 30 days. Fifteen percent (90) of Black youths reported using cigarettes and 17% (107) reported using lean or krokodil in the past 30 days. Only 6% reported using ecstasy and 3% self-reported using crack or cocaine in the last 30 days. Of all youth, only 31% reported using a condom during their last time of having sexual intercourse. Lastly, 9% of the sample reported having sex while on alcohol/drugs without condoms.

Table 2 presents a comparison of the sample characteristics and sociocultural factors in relation to drug use among Black youth. A Chi-square test of independence was performed to examine the relationship between age and drug use. In regard to different age groups, those between the ages of 19 and 22 (χ^2^(1) = 19.81, *p* < 0.001) as well as between 15 and 18 (χ^2^(1) = 18.22, *p* < 0.001) were all significantly associated with drug use. In regard to different sexual orientation groups, those who self-identified as gay (χ^2^(1) = 14.58, *p* < 0.001) and bisexual (χ^2^(1) = 14.55, *p* < 0.001) were associated with drug use. Condom use was associated with drug use to a greater extent in the case of those who said no to condom use compared to those who said yes (χ^2^(1) = 5.29, *p* < 0.021).

Table 3 presents a comparison of the sample characteristics and sociocultural factors in relation to sex under the influence of drugs or alcohol among Black youth. It was observed that there was a statistically significant association between those who self-identified as being male and drug use (χ^2^(1) = 7.98, *p* < 0.05). A Chi-square test of independence was performed to examine the relationship between age and sex while on drugs, which was not statistically significant. Having sex in exchange for drugs was associated with sex under the influence of drugs or alcohol (χ^2^(1) = 33.78, *p* < 0.001). Having sex in exchange for something other than drugs was associated with sex under the influence of drugs or during alcohol use (χ^2^(1) = 38.75, *p* < 0.001).

### 3.2. Drug Use

As presented in Table 4, the overall model was statistically significant. An increase in parent–child sexual health communication decreased the prevalence of drug use among Black youth (PR: 0.01, 95% CI: 0.00, 0.02). The prevalence of drug use decreased with an increase in parental bonding (PR: 0.14, 95% CI: 0.05, 0.23). Lower levels of peer pressure decreased the prevalence of drug use in youth (PR: 0.39, 95% CI: 0.28,0.49). An increase in condom use decreased the prevalence of drug use (PR = 0.52, 95% CI: 0.28, 0.95). Drug prevalence was higher among males than females (PR = 0.74, 95% CI: 0.61, 0.90).

### 3.3. Sex under the Influence of Drugs or Alcohol

As presented in Table 5, the overall model was statistically significant. Peer pressure was statistically significant and associated with sex under the influence of drugs/alcohol (OR = 0.17, 95% CI: 0.07, 0.37). Condom use was statistically significant and associated with having sex under the influence of drugs/alcohol (OR = 5.99, 95% CI: 1.32, 7.16). Those who identified as being male were more likely to have sex under the influence of drugs/alcohol when compared to females (OR = 0.16, 95% CI: 0.04, 0.35).

## 4. Discussion

This study examined the relationship between parent–child sexual health communication, parent–child relationships, peer pressure, and other contextual factors regarding drug use and sexual risk-taking among Black youth and young adults, which reflects the micro- and mesosystem levels of ecodevelopmental theory. The regional data reflected findings regarding adolescents’ engagement in risky behaviors, which were a cause of concern. Our findings indicated that an increase in parent–child sexual communication, parental bonding, and lower levels of peer pressure decreased the prevalence rate of drug use among Black youth. These findings are consistent with the prior literature that parental factors can be protective for Black youth [21,24]. Our results also indicated that youth who did not experience peer pressure were less likely to have sex while under the influence of alcohol or drugs, which suggests the effect of the mesosystem on Black youth and young adults in this sample. This finding aligned with previous research that examined the role of peer pressure and its influence on adolescent’s risk behaviors [46,47,48]. Understanding how parental processes influence substance use and sexual behavior among adolescents is crucial in prevention and intervention research. Further, this study contributes significantly to the literature on substance use among Black adolescents, especially since this is one of the few studies that has examined substance use and co-occurring risk behaviors, i.e., sex while under the influence of alcohol and drugs, and their association with protective factors, i.e., parental bonding and parent–child sexual health communication.

### 4.1. Parent-Child Sexual Communication

Surprisingly, our findings revealed that parent–child sexual health communication was associated with drug use but had no relationship with sex while using substances reflecting nuances at the microsystem level. This can be due to parents focusing more on sex and not discussing certain risks associated with the co-occurrence of sex and drug use, which may leave adolescents unaware of the potential harms of using drugs or result in them viewing drug use as less risky than sex. Adolescents in the study who had had an earlier sexual debut (i.e., before the age of 12) were also more likely to use substances while having sex. This could be due to a lack of prevention efforts targeted toward younger adolescents who have sex, leaving them at greater risk of engaging in risky behaviors, such as substance use [49,50].

### 4.2. Parental Bonding

Our findings indicated that the prevalence of drug use decreased with higher levels of parental bonding among Black youth, which was a notable protective effect. First, the empirical literature has reflected the positive effects of parents’ roles on their children and their decision making regarding sexual behaviors [51,52]. This has also been consistent with the literature that reported on parental bonding and how it may result in the youth feeling supported by their parents, which can influence the youth to make more health-conscious decisions that protect against negative health outcomes associated with substance use among the youth [19,53,54]. Further, this is an important finding because substance use, misuse and abuse have been identified as key risk factors for engagement in risky sexual behavior that often leads to STIs and HIV. Youth who have strained parental bonds are more likely to engage in risky behaviors, such as having sex in exchange for drugs or something other than drugs. Enhancing the parent–child relationship is key in drug use and sexual health prevention and intervention work, as it can serve as a protective mechanism against youth engagement in risky behaviors.

### 4.3. Peer Pressure

Our results indicated that lower peer pressure decreased the prevalence of drug use. We also found that youth who did not experience peer pressure were less likely to have sex while on drugs. This can be due to positive peer pressure, and adolescents discussing the importance of delaying alcohol or drug use and sex [55]. Furthermore, this can be that youth are involved in activities that promote healthy behaviors such as sports, which reduces boredom that leads to drug use [56]. In addition, parents may also play a critical role in reducing peer pressure that influences drug use and sexual debut. Parent communication has been found to delay sex among youth. The quality of the parent–child relationship, such as parental support, involvement, and general communication has been found to delay alcohol use and influence their subsequent level of alcohol use [56]. Developing interventions that empower youth to make healthy decisions may help increase positive peer pressure.

### 4.4. Gender Differences

Our results indicated that the prevalence of drug use was higher among males than females. In addition, we also found that males were more likely to report having sex while on drugs or alcohol than females. This is consistent with the prior literature that adolescent males generally engage in higher rates of alcohol use compared to girls [57] and are more likely to report having sexual intercourse [51]. One reason that findings may differ for boys and girls may be due to masculine norms. Prior research has found that sexual activity, engaging in heavy drinking and the ability to consume large amounts of alcohol are an expression of masculinity [58]. Our findings suggest that research should develop culturally relevant alcohol reduction interventions and prevention efforts for both Black girls and boys, but especially boys. This will allow researchers and practitioners to help better understand Black adolescent boys who are engaging in drug and alcohol use.

### 4.5. Limitations

Although this study contributes significantly to the literature on co-occurring risk behaviors among Black adolescents, it is not without its limitations. First, we were unable to establish the causality and directionality of the associations found due to the cross-sectional design of the study. Additionally, the study findings may not be generalizable to all youth, especially those outside the Midwest. Residual confounding may also be a problem with some of the findings. We encourage future researchers to conduct longitudinal studies to examine the impact of parental communication and bonding on adolescent behaviors over time. Second, we measured parent–child sexual health communication based on adolescent reports. Despite the quality of the survey questions, there remained the potential for some degree of under- or over reporting bias. Future research should also collect communication measurements from parents in order to assess whether their reports are consistent. Investigating trends in youth sexual behavior and substance use by gender will allow us to determine where to focus prevention and intervention efforts that can center the unique and important role that parents and parent–child communication play in the lives of their children.

### 4.6. Implications

Our study contributes to the literature on substance use by providing insight into how parents communicate about the intersection of sex and drugs and their impact on youth behaviors. While parents may be unaware of how to discuss co-occurring risk behaviors such as having sex while using substances, we encourage researchers and prevention specialists to incorporate more parent–child programming that highlights the parent–child relationship, especially in the context of Black families, and emphasizes the discussion on the intersection of substance use and sexual health communication. Such strategies can include incorporating parent–child sexual health and substance use communication discussions during family-based initiatives in urban communities that are targeted for Black families. Strengthening Families is an evidence-based intervention that seeks to improve the family–child relationship [59,60]. Incorporating more discussion about the importance of sexual health and substance use communication with children can be beneficial in improving skills. In addition, parent–child interventions may encourage parents to be direct in discussions surrounding sex or drugs. Opara et al. [19] found that parents of youth who lived in an urban neighborhood, used their environment, which had visible effects of drug use and abuse due to multiple drug epidemics plaguing their community, as a prompt for discussions around drug use specifically. Other studies have suggested that parent–child communication about drug use and sexual health should avoid being too directive or demanding because of the possibility of adolescents doing the opposite or having been already engaged in those behaviors [61,62]. It is ideal for parents to begin relatable discussions surrounding sex and drug use concurrently and earlier with their children, in order to prepare their children to resist negative behaviors associated with drug use and sex, especially paying special attention to environment and how surroundings can normalize certain behaviors in youth.

Our findings revealed the importance of the parent–child relationship and its influence on substance use and sexual behavior. Our findings also indicated that peer pressure contributes to substance use and sex under the influence of drugs. We recommend that prevention and intervention efforts align with global and domestic standards such as the United Nations Convention on the Rights of the Child, who declared adolescents across the globe “have the human right to receive sexual and reproductive health care” (U.N. Committee on the Rights of the Child). This recommendation is aligned with other organizations such as the World Health Organization, the Society for Adolescent Health and Medicine and the American Academy of Pediatrics [63]. Specifically, health professionals should explicitly recommend the importance of parent–child sexual health communication during each visit to promote communication prior to sexual debut and substance use for Black adolescents. Therefore, this presents a crucial implication regarding the investment of resources in developing an intervention to reduce peer pressure among youth and empowering them to make the best decisions for themselves. It is important to note that Black families in urban settings face unique ecological factors that may affect family communication and bonding quality due to their socioeconomic status, neighborhood factors, discrimination, and racism. Future research should also seek to understand how Black parents can be supported in healing from their stressful experiences while also effectively communicating with their children. Specifically, culturally adapting and implementing effective intervention that are also brief could improve parent–child communication while also effectively monitoring adolescent behavior, which may result in delayed sexual debut [64]. Given the likelihood of alcohol and drug use during sex with this population, aspects of effective interventions need to also reflect the propensity of ways to be more responsive, especially for Black boys.

Qualitative and mixed methods research may offer other empirical approaches the opportunity to explore sexual health communication between Black parents and their children in greater depth. It also may be a way to describe in detail what their conversations entail and why they exclude or include certain information. Moreover, individual factors such as the parent and adolescents’ age, gender, parents’ attitudes toward and beliefs about sexual behavior, parent and child psychopathology and strengths, parent comfort and knowledge, parents’ socioeconomic status, and parents’ views on drug education and sexual health should be explored in future research. In addition, given the unique experiences that Black parents and their children may have in urban settings, it may be worthwhile for future research to explore the insidious influence that structural (poverty and racism) and systemic (involvement with other systems such as child welfare and/or juvenile/criminal justice) factors have on their life quality. However, there should also be acknowledgement that although these factors may exist and persist, there are pertinent strengths and protective factors Black families possess that need to be considered in future work such as personal agency, positive parenting and parental bonding [51,52]. The implementation of both a community-driven and science-based public health approach to urban prevention and intervention practice establishes the capacity for reducing the incidence of co-occurring sex and drug/alcohol misuse and creating the conditions that promote wellness and health equity for Black youth and their parents.

## Figures and Tables

**Table 1 ijerph-18-05170-t001:** Sample characteristics (*N* = 636).

Variable	Frequency (%)		Total Response (%)
Gender	Yes	No	
Male			290 (45%)
Female			346 (54%)
Drug use			
Marijuana use	372 (60)	250 (40)	622
Alcohol use	315 (49.5)	321 (50)	636
Ecstasy use	37 (6)	598 (94)	636
Cigarette use	90 (15)	543 (85)	636
Lean or krokodil	107 (17)	515 (83)	622
Crack or cocaine use	15 (3)	602 (97)	622
Sexual behaviors			
Sexual debut	203 (60)	137 (40)	340
Sex while on alcohol/drugs without condoms	31 (9)	331 (91)	362
Use of condom during last sexual intercourse	110 (31)	245 (69)	355

Note. The mean age for the study sample is 16, and the SD is 1.41. The median age for sexual debut is 14.

**Table 2 ijerph-18-05170-t002:** Comparison of Chi-square test between drug use and sociocultural variables (*N* = 613).

Variable	Drug Use	No Drug Use	χ^2^	*p*-Value
Gender				
Male	189	87	0.042	0.828
Female (reference)	109	228		
Age				
12–14 (reference)	20	95		
15–18	137	273	18.22	0.001
19–22	39	48	19.18	0.001
Government assistance			2.95	0.090
Yes (reference)	39	154		
No	110	304		
Sexual orientation				
Heterosexual (reference)	141	324		
Gay	13	11	14.58	0.012
Bisexual	27	31	14.55	0.012
Pansexual	3	4		
Transgender	0	3		
Other	3	10		
Condom use				
Yes (reference)	151	86	5.29	0.021
No	52	53		
Having sex in exchange for drugs				
Yes (reference)	27	14	1.309	0.245
No	173	127		
Having sex in exchange for something other than drugs				
Yes (reference)	10	5	0.529	0.467
No	164	123		

**Table 3 ijerph-18-05170-t003:** Comparison of Chi-square test between sex while on drugs or alcohol and sociocultural variables.

Variable	Sex on Drugs or Alcohol	No Sex on Drugs or Alcohol	χ^2^	*p*-Value
Gender				
Male	25	187	7.98	0.005
Female (reference)	5	142		
Age				
12–14 (reference)	3	27		
15–18	24	239	1.56	0.458
19–22	3	63	1.77	0.395
Government assistance				
Yes (reference)	24	254	0.069	0.792
No	6	72		
Sexual orientation				
Heterosexual (reference)	23	241		
Gay	1	12	1.55	0.906
Bisexual	2	37	2.13	0.830
Pansexual	0	6		
Transgender	0	1		
Other	1	6		
Condom use				
Yes (reference)	223	102	28.59	0.593
No	8	22		
Having sex in exchange for drugs				
Yes (reference)	19	9	33.78	0.001
No	34	247		
Having sex in exchange for something other than drugs				
Yes (reference)	24	5	38.75	0.001
No	164	284		

**Table 4 ijerph-18-05170-t004:** Estimated prevalence ratio for drug use among Black youth (*N* = 340).

	Prevalence Ratios (PR)	95% CI
Parent–child sexual health communication	0.01 **	(0.00, 0.02)
Parental bonding	0.14 **	(−0.21, −0.03)
Peer pressure	0.39 ***	(0.28, 0.49)
Condom use (yes, reference)	0.52 *	(0.28, 0.95)
Sexual orientation		
Heterosexual		
Gay	1.16	(0.74, 1.80)
Bisexual	1.11	(0.87, 1.40)
Other	0.48	(0.12, 1.90)
Pansexual	1.55	(0.18, 12.74)
Age		
12–14 (reference)		
15–18	1.28	(0.88, 1.87)
19–22	1.24	(0.81, 1.89)
Gender (female)	0.74 **	(0.61, 0.90)
Government assistance (yes, reference)	0.05	(−2.03, 0.21)
Having sex in exchange for drugs (yes, reference)	0.91	(0.60, 1.37)
Having sex in exchange for something other than drugs (yes, reference)	1.38	(0.90, 2.12)

Note. * Adjusted for age, sex, sexual orientation, government assistance, and gender. * *p* < 0.05, ** *p* < 0.01, *** *p* < 0.001.

**Table 5 ijerph-18-05170-t005:** Multivariable analysis sex on drugs/alcohol (*N* = 340).

	OR	SE	95% CI
Parent–child sexual health communication	0.99	0.24	(0.94, 1.04)
Parental bonding	0.73	0.16	(0.48, 1.12)
Peer pressure	0.17 ***	0.69	(0.07, 0.37)
Condom use (yes, reference)	5.99 *	4.11	(1.32, 7.16)
Sexual orientation			
Heterosexual			
Gay	4.67	5.99	(0.37, 10.78)
Bisexual	0.72	0.07	(0.09, 5.32)
Other			
Pansexual			
Age			
12–14 (reference)			
15–18	0.74	0.13	(0.06, 8.32)
19–22	0.15	0.21	(0.01, 2.56)
Gender (female)	0.16 **	0.13	(0.04, 0.35)
Government assistance (yes, reference)	1.24	0.59	(0.48, 3.20)
Having sex in exchange for drugs (yes, reference)	0.27	0.24	(0.05, 1.54)
Having sex in exchange for something other than drugs (yes, reference)	1.32	1.71	(0.10,16.70)

* *p* < 0.05, ** *p* < 0.01, *** *p* < 0.001.

## Data Availability

Data sharing is not applicable to this article.

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
