# Peer review of "Associations between Parent–Child Communication on Sexual Health and Drug Use and Use of Drugs during Sex among Urban Black Youth"

_ijerph, 2021, doi:10.3390/ijerph18105170_

Round 1

Reviewer 1 Report

Boyd et al have examined the role of parent-youth bonding and sexual health communication and investigated whether contextual factors contribute to or prevent substance use and sex while on drugs among this population in an urban context. Authors have presented their results clearly and  suggested that parental bonding plays a critical role in youth using drugs while living in urban environments. 

They recruited a population whose household income was between $24,049 and $35,946, which was below the city average of approximately $43,000.

It is understood that people with low income have a lot of challenges in their daily life and may not be emotionally strong to focus on their children. I am just wondering what are the strategies that can be implemented to overcome this challenge as we can't improve the income of the parents. 

Authors may suggest few strategies  to educate the parents about talking to their kids regarding healthy sex and other factors.

Author Response

Reviewer Comments

Responses

Reviewer 1

It is understood that people with low income have a lot of challenges in their daily life and may not be emotionally strong to focus on their children. I am just wondering what are the strategies that can be implemented to overcome this challenge as we can't improve the income of the parents. 

The authors welcome your comment.  The authors provided strategies that can be implemented to overcome this challenge as we can't improve the income of the parents on page 13.

“We recommend that prevention and intervention efforts align with global and domestic standards like the The United Nations Convention on the Rights of the Child who declared adolescents across the globe “have the human right to receive sexual and reproductive health care” (U.N. Committee on the Rights of the Child). This recommendation is aligned with others organizations like the World Health Organization, Society for Adolescent Health and Medicine and the American Academy of Pediatrics (Society for Adolescent Health and Medicine, 2014). Specifically, health professionals should explicitly recommend the importance of parent–child sexual health communication during each visit to promote communication prior to sexual debut and substance use for Black adolescents.”

“Specifically, culturally adapting and implementing effective intervention that are also brief could improve parent–child communication while also effectively monitoring adolescent behavior, which may result in delayed sexual debut (Santa Maria et al., 2017). Given the likelihood of alcohol and drug use during sex with this population, aspects of effective interventions need to also reflect the propensity of ways to be more responsive, especially for Black boys.”

“In addition, given the unique experiences that Black parents and their children may have in urban settings, it may be worthwhile for future research to explore the insidious influence of structural (poverty and racism) and systemic (involvement with other systems like child welfare and/or juvenile/criminal justice) factors have on their life quality. However, there should also be acknowledgement that although these factor may exist and persist, there are pertinent strengths and protective factors Black families possess that need to be considered in future work like personal agency, positive parenting and parental bonding (Boyd et al., 2020; Boyd et al., 2021; Quinn et al., 2021). The implementation of both a community driven and science-based public health approach to urban prevention and intervention practice establishes the capacity for reducing the incidence of co-occurring sex and drug/alcohol misuse and creating the conditions that promote wellness and health equity for Black youth and their parents.”

Authors may suggest few strategies  to educate the parents about talking to their kids regarding healthy sex and other factors.

The authors welcome your comment.  The authors provided strategies to educate the parents about talking to their kids regarding healthy sex and other factors on pages 12-13. Please see below:

 “Such strategies can include incorporating parent-child sexual health and substance use communication discussions during family-based initiatives in urban communities that are targeted for Black families. Strengthening Families is a evidence-based intervention that seeks to improve the family-child relationship (Kumpfer et al., 1989; 2020). Incorporating more discussion about the importance of sexual health and substance use communication with children can be beneficial in improving skills. In addition, parent-child interventions may encourage parents to be direct in discussions surrounding sex or drugs. Opara et al (2019) found that parents of youth who lived in an urban neighborhood, used their environment, which had visible effects of drug use and abuse due to multiple drug epidemics plaguing their cmmunity,  as a prompt for discussions around drug use specifically. Other studies have suggested that parent‐child communication about drug use and sexual health should avoid being too directive or demanding because of the possibility of adolescents doing the opposite or having been already engaged in those behaviors (Bonafide et al., 2020; Ennet et al.,2001 ). It is ideal for parents to begin relatable discussions surrounding sex and drug use concurrently and earlier with their children, in order to prepare their children to resist negative behaviors associated with drug use and sex especially paying special attention to environment and how surroundings can normalize certain behaviors in youth”.

Reviewer 2 Report

The paper show interesting data, but could be improved.
In young people, it would be very important to know not only
if they have consumed or not, but also the frequency of consumption,
since exploratory consumption is frequent at these ages.
Establishing a single category in consumption causes essential
information to be lost. Another element to take into account is the polydrug use
of substances, which is a risk factor and should be included
in the study as one more variable.

Author Response

Reviewer 2 Comments

Authors Response

The paper show interesting data but could be improved. In young people, it would be very important to know not only if they have consumed or not, but also the frequency of consumption, since exploratory consumption is frequent at these ages. Establishing a single category in consumption causes essential information to be lost. Another element to take into account is the polydrug use of substances, which is a risk factor and should be included in the study as one more variable.

·       The authors welcome your comment. The authors agree with your statement.  The authors updated the frequencies of consumption in Table 1 (sample characteristics). The authors also updated the results with frequencies on pages 6-7.

·       The authors agreed with the reviewer second comment about establishing a single category in consumption causes essential to be lost. The authors went back and coded the Drug Use variable to create a count variable, please see page 4. The authors also conducted an appropriate analysis for a count variable. The authors conducted a Poisson regression analysis with robust errors to account for Drug Use. The authors reported the prevalence ratios (PR), which is more appropriate for this analysis. Please see the updated Statistical Analysis plan on pages 5-6. Please see the updated results on page 9 and the updated Table. 4 on page 10.  All analysis was done in STATA 17 and guided by literature. Please see below for references.

Cash RE, Anderson SE, Lancaster KE, Lu B, Rivard MK, Camargo Jr CA, Panchal AR. Comparing the prevalence of poor sleep and stress metrics in basic versus advanced life support emergency medical services personnel. Prehospital Emergency Care. 2020 Sep 2;24(5):644-56.

Barros AJ, Hirakata VN. Alternatives for logistic regression in cross-sectional studies: an empirical comparison of models that directly estimate the prevalence ratio. BMC medical research methodology. 2003 Dec;3(1):1-3.

StataCorp. Stata survey data reference manual release 15. College Station, TX: Stata Press; 2017.

Round 2

Reviewer 2 Report

In the last version, the paper could be published